# A Novel De Novo Missense Mutation in *KIF1A* Associated with Young-Onset Upper-Limb Amyotrophic Lateral Sclerosis

**DOI:** 10.3390/ijms25158170

**Published:** 2024-07-26

**Authors:** Emilien Bernard, Florent Cluse, Adrien Bohic, Marc Hermier, Cédric Raoul, Pascal Leblanc, Claire Guissart

**Affiliations:** 1Lyon ALS Reference Center, Hôpital Neurologique Pierre Wertheimer, Hospices Civils de Lyon, Université de Lyon, 59 Boulevard Pinel, 69677 Bron, France; florent.cluse@chu-lyon.fr (F.C.); adrien.bohic@chu-lyon.fr (A.B.); 2Institut NeuroMyoGène, CNRS UMR5310, INSERM U1217, Faculté de Médecine Rockefeller, Université Claude Bernard Lyon I, 8 Avenue Rockefeller, CEDEX 08, 69373 Lyon, France; pascal.leblanc@univ-lyon1.fr; 3Department of Neuroradiology, Hôpital Neurologique Pierre Wertheimer, Hospices Civils de Lyon, Université de Lyon, 59 Boulevard Pinel, 69677 Bron, France; marc.hermier@chu-lyon.fr; 4INM, Université de Montpellier, INSERM, CNRS, 34295 Montpellier, France; cedric.raoul@inserm.fr (C.R.); claire.guissart@chu-nimes.fr (C.G.); 5ALS Reference Center, Université de Montpellier, CHU Montpellier, 34295 Montpellier, France; 6GCS AURAGEN, 69003 Lyon, France; 7Service de Biochimie et Biologie Moléculaire, CHU Nîmes, Université de Montpellier, Place du Professeur Robert Debré, 30029 Nîmes, France

**Keywords:** amyotrophic lateral sclerosis, genome sequencing, *KIF1A*, de novo mutation

## Abstract

We investigate the etiology of amyotrophic lateral sclerosis (ALS) in a 35-year-old woman presenting with progressive weakness in her left upper limb. Prior to sequencing, a comprehensive neurological work-up was performed, including neurological examination, electrophysiology, biomarker assessment, and brain and spinal cord MRI. Six months before evaluation, the patient experienced weakness and atrophy in her left hand, accompanied by brisk reflexes and Hoffman sign in the same arm. Electroneuromyography revealed lower motor neuron involvement in three body regions. Neurofilament light chains were elevated in her cerebrospinal fluid. Brain imaging showed asymmetrical T2 hyperintensity of the corticospinal tracts and T2 linear hypointensity of the precentral gyri. Trio genome sequencing identified a likely pathogenic de novo variant in the *KIF1A* gene (NM_001244008.2): c.574A>G, p.(Ile192Val). Pathogenic variants in *KIF1A* have been associated with a wide range of neurological manifestations called KIF1A-associated neurological diseases (KAND). This report describes a likely pathogenic de novo variant in *KIF1A* associated with ALS, expanding the phenotypic spectrum of KAND and our understanding of the pathophysiology of ALS.

## 1. Introduction

Amyotrophic lateral sclerosis (ALS) is a neurodegenerative disorder characterized by the degeneration of upper and lower motor neurons leading to death a median three years after onset. However, a considerable variability in the rate of progression of disease can be observed [1], ranging from slow to acute and dramatic forms. With the growing number of genes found to be associated with the disease, which is approaching 40 [2], and the easier access to sequencing technologies, a pathogenic mutation is now identified in 70% of familial ALS cases [2] and up to 10% of sporadic cases [3]. However, due to the typical late onset of the disease, interpreting rare gene variants in ALS can be challenging. This difficulty arises because there is often no opportunity to observe the segregation of the suspected variant within the family, either because the parents of the proband have passed away or because other family members are not available for testing. This situation can prevent the discovery of de novo variants.

The kinesin family member 1A gene (*KIF1A*) encodes a protein that is responsible for the ATP-dependent anterograde axonal transport of synaptic vesicle precursors along the axonal and dendritic microtubules [4]. Initially, biallelic pathogenic mutations in *KIF1A* were linked to two phenotypes: a recessive form of hereditary spastic paraplegia (HSP) called spastic paraplegia-30 (SPG30), and the hereditary sensory and autonomic neuropathy type II. Further reports have shown that monoallelic variants in *KIF1A* are associated with a wide range of neurological manifestations, which are collectively referred to as KIF1A-associated neurological diseases (KANDs). These neurological manifestations include pure or complex HSP, Rett-like syndrome, ataxia, epilepsy, progressive encephalopathy with edema, hypsarrhythmia and optic atrophy (PEHO) syndrome, or optic atrophy [5].

In this report, we conducted a genome sequencing of a trio consisting of a young-onset ALS patient and her parents. This analysis revealed a likely disease-causing new genetic variant in *KIF1A* that arose spontaneously in the patient. This finding broadens the range of phenotypes associated with KAND and supports the suggested link [6,7] between *KIF1A* and the pathophysiology of ALS.

## 2. Case Description

The propositus was a 35-year-old woman, the eldest of two sisters. Her parents did not present any neurological disorder (Figure 1A). She had no significant medical history. She reported a 6-month history of distal left upper limb weakness preceded by cramps, without any sensory disturbance nor pain. Clinical examination found weakness and atrophy of hand muscles (Medical Research Council 2–3/5 in distal and 4/5 in proximal left upper limb muscles). Rare fasciculations were observed on both shoulders and thighs. Deep tendon reflexes were brisk in the four limbs, with left Hoffman sign, indicating upper motor neuron involvement. Needle electromyography found chronic and active denervation in proximal and distal muscles of cervical, thoracic and lumbar regions, with normal sensory conductions. Elevated NfL was found in the CSF (1614 ng/L; normal value < 750 ng/L). Spinal cord MRI was unremarkable. Brain MRI was evocative of motor neuron disease, displaying asymmetrical T2 hyperintensity of corticospinal tracts and T2 linear hypointensity of precentral gyri, more pronounced in the right hemisphere (Figure 2). Diagnosis of ALS was made according to El Escorial (definite) and Gold Coast criteria. Riluzole was initiated and the patient enrolled in a clinical trial. After 18 months of follow-up, disease progression was rather slow: left arm weakness was aggravated but only minor proximal right upper limb weakness developed, without impairment in walking, respiratory or bulbar symptoms.

The absence of the pathogenic *C9ORF72* hexanucleotide repeat expansion in our patient was confirmed prior to initiating whole-genome sequencing analysis. Trio genome sequencing revealed a novel heterozygous de novo missense variant in *KIF1A* in the proband (transcript ID ref: NM_001244008.2: c.574A>G, p.(Ile192Val)). The variant was confirmed by Sanger sequencing, ensuring the accuracy of the genomic findings (Figure 1B). This variant is not in public databases (e.g., gnomAD v4) and is located in the kinesin motor domain, which is known to be prone to mutations. In silico tools, such as Polyphen2, ClinPred and Mistic, predicted that the amino acid change (p.Ile192Val) is damaging. Isoleucine 192 is highly conserved across all eukaryotes, including plants (Figure 1C). According to the American College of Medical Genetics (ACMG) criteria [8], the *KIF1A* variant was classified as likely pathogenic (class 4). This classification was based on the following ACMG criteria: PS2 (de novo variant), PM1 (located in a mutational hot spot and/or critical and well-established functional domain), and PM2 (absent from controls in population databases). None of the other variants identified in this analysis exhibited sufficient evidence to support their involvement in the disease.

## 3. Discussion

*KIF1A* has recently been identified as a candidate gene associated with ALS in two separate cohorts from Norway [6] and China [7]. In the Norwegian cohort, rare damaging variants of *KIF1A* were found in 3/279 ALS patients [6] while, in the Chinese cohort, 10/941 ALS patients had these variants [7]. However, these variants were considered to have unknown significance according to the ACMG criteria, and therefore they were deemed unsuitable for definitive genetic counseling for these patients. The present study presents the evidence of a probably pathogenic variant, classified as ACMG class 4, in *KIF1A* that is associated with ALS. The propositus exhibited classical spinal upper-limb onset ALS associated with typical electrophysiological and MRI features. The only unusual aspects were the young age at onset and the relatively slow progression of the disease. The levels of NfL in the CSF, which are known to be correlated with prognosis [9], were found to be three times lower than the mean levels observed in ALS patients [10].

*KIF1A* can be added to the list of ALS-causing genes that affect cytoskeletal dynamics, joining *ALS2*, *DCTN1* and *KIF5A* [11]. However, it is important to note that the *KIF1A* variant described here (c.574A>G, p.(Ile192Val)) is located in the motor domain of the kinesin protein. This contrasts with previously identified ALS-associated variants in *KIF5A*, which are found in the cargo-binding domain, leading to the formation of cytoplasmic aggregates, inclusions of mitochondria, and synaptic vesicles that are toxic to motor neurons through a gain-of-function mechanism [12,13,14], whereas spastic paraplegia and Charcot–Marie–Tooth-associated variants are located in the motor domain. This discrepancy suggests that while mutations in the motor domain of kinesin proteins can lead to a variety of neurological diseases, the specific domain affected may influence the phenotype. While the connection between *KIF1A* mutations and the characteristic extranuclear aggregates of TDP-43 in ALS is not yet understood, other kinesins, such as DCTN1, have been reported to disrupt the dynamics of stress granules and promote the formation of TDP-43 cytoplasmic aggregation in cultured cells [15]. 

Further supporting the pathogenicity of the *KIF1A* c.574A>G variant, the Human Gene Mutation Database (HGMD) reports a disease-causing mutation in the same codon (c.575 T>G, p.(Ile192Ser)), associated with intellectual disability. The REVEL score for p.(Ile192Ser) is >0.8, indicating a high likelihood of pathogenicity, while the score for p.(Ile192Val) is 0.4. Despite these differing predictions, the strong evolutionary conservation of isoleucine at position 192 across species and the damaging predictions from multiple in silico tools support the variant’s likely pathogenic nature. Furthermore, isoleucine 192 is situated within the alpha-5 helix of the KIF1A catalytic core, a critical domain known for its highly processive movement [16]. This domain confers KIF1A with the ability to move more than five times as processively as dimeric conventional kinesin [17]. 

Numerous mutations have been identified in the motor domain of KIF1A as a cause of KAND, with both loss-of-function and gain-of-function effects on axonal transport [18]. The elucidation of the mechanisms underlying neurodegeneration and the association with ALS pathological markers, including TDP-43 aggregates, can be further advanced through the future development of models such as motor neurons derived from patient induced pluripotent stem cells or Drosophila. This approach has already been successfully employed in the case of *KIF5A* [12,13].

As clinical trials in the nano-rare disease KAND are currently underway with antisense oligonucleotides targeting *KIF1A* [19], the present report emphasizes the importance of systematic trio sequencing in sporadic ALS. In the event of de novo mutations, this sequencing approach can upgrade the level of pathogenicity of suspected variants from unknown significance to probably pathogenic.

## 4. Methods

The patient and her parents provided informed written consent for genome sequencing and publication of findings. Peripheral venous blood was collected from the family trio. Whole-genome sequencing was performed following the recommendations of 2025 French genomic medicine initiative. Genomic DNA extracted from whole blood was sequenced according to standard procedures for a PCR-Free genome on a NovaSeq6000 instrument (Illumina, San Diego, CA, USA). Sequencing data were aligned to the GRCh38p13 full assembly using Burrows–Wheeler Aligner version 0.7+. The sequencing achieved an average target depth of approximately 42×, with over 98% of the genome covered at a depth greater than 20×. Variants were called by several algorithms, including GATK4+, Bcftools1.10+, Manta1.6+, and CNVnator0.4+, and annotated using the variant effect predictor. Variant prioritization involved several key filtering steps. Initially, variants with a minor allele frequency greater than 1% in gnomAD were excluded. Subsequently, we filtered variants to retain only those affecting coding sequences or splice sites and predicted to be deleterious by multiple in silico tools (e.g., for missense variants: PolyPhen-2, ClinPred, REVEL; for splicing variants: SpliceAI). Variants previously documented as pathogenic or likely pathogenic in ClinVar, as well as de novo or compound heterozygous variants, were prioritized. Finally, variants were evaluated in relation to the patient’s phenotype and established gene-disease associations. Commercially available ELISA kits were used for the measurement of neurofilament light chain (NfL) concentration in cerebrospinal fluid (CSF; Uman Diagnostics, Umeå, Sweden). MRI was performed on a clinical 3.0-T Philips Ingenia system (Philips Healthcare, Amsterdam, The Netherlands).

## Figures and Tables

**Figure 1 ijms-25-08170-f001:**
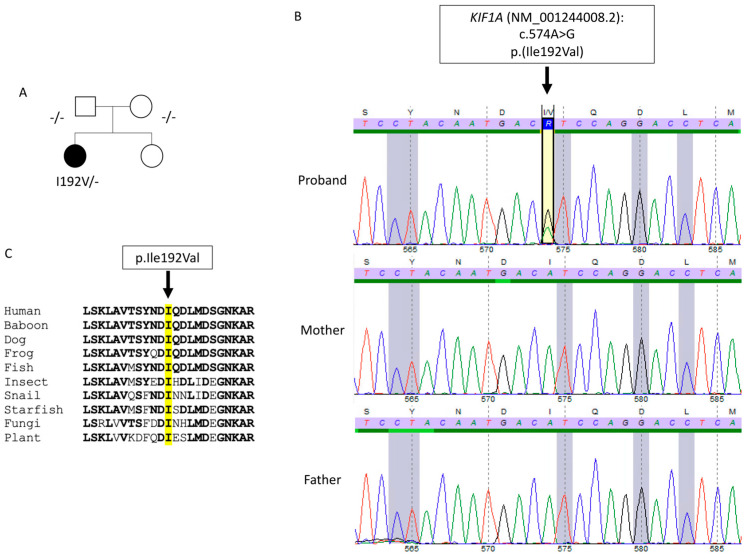
Identification of *KIF1A* mutation. (**A**) Pedigree of the affected family. (**B**) Sanger sequencing of the c.574A>G mutation for the proband and her parents. The de novo heterozygous c.574A>G mutation in the *KIF1A* gene is indicated by an arrow. (**C**) Amino acid sequence comparison of the KIF1A motor domain and orthologous proteins from various species. Identical amino acids to the human *KIF1A* sequence are shown in bold. The missense mutation is indicated by an arrow on top of the mutated amino acid Isoleucine (I) 192 highlighted in yellow.

**Figure 2 ijms-25-08170-f002:**
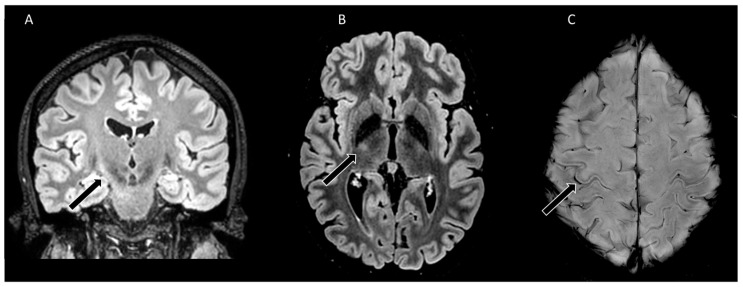
Brain MRI of a 35-year-old woman with ALS harboring a de novo probably pathogenic c.574A>G *KIF1A* mutation. Coronal (**A**) and axial (**B**) FLAIR sequences revealed asymmetrical hypersignal of the corticospinal tracts (black arrow indicating the right corticospinal tract). Axial SWI sequence (**C**) displayed linear hyposignal of the right precentral gyrus (black arrow). Abbreviations: FLAIR: Fluid-attenuated inversion recovery, SWI: Susceptibility weighted imaging.

## Data Availability

The original contributions presented in the study are included in the article; further inquiries can be directed to the corresponding author.

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
