# Peer review of "A Novel De Novo Missense Mutation in KIF1A Associated with Young-Onset Upper-Limb Amyotrophic Lateral Sclerosis"

_ijms, 2024, doi:10.3390/ijms25158170_

Round 1

Reviewer 1 Report

Comments and Suggestions for Authors

The manuscript by Bernard et al presents an interesting case in which KIF1A may be associated with an ALS phenotype. Although this perspective is intriguing, and in line with previous literature, I am not convinced that the data reported is sufficient to support the claims of the paper. In particular, the analysis of sequencing data is obscure, with many key steps in variant calling analysis and prioritization left undescribed: how to combine the results of multiple variant caller? Why to focus on this specific variant among the many variants potentially contributing to the ALS phenotype? As things stand, the authors identified the co-occurrence of the KIF1A variant with an ALS phenotype: this is far from a place where we could claim that this variant is indeed causative of the phenotype. The “in-house” method to prioritize variants is not acceptable in a peer review process. Finally, there is no effort in validating the variant via golden standard methods such as Sanger sequencing. Altogether, I am not convinced that the claim made in the paper is sufficiently demonstrated nor it could be demonstrated with additional revisions. Therefore, I am inclined to reject the submission.

Line by line comments

L18: I would remove words such as "Objectives", "Methods" etc and opt for a narrative paragraph instead

L40: “number of genes”

L40: the number of genes (40) requires a citation

L44 unclear, rephrase “absence of proven segregation”

L49 define HSP

L54 define PEHO

L57 downplay the sentence. “KIF1A variant found only in the patient”. Although unlikely, you can’t rule out a sequencing error unless a Sanger sequencing is performed.

L61 why the sister was not included in the sequenced sample?

L70 refrain from using acronyms without defining them on fist occurrence

L86 there is no mention of the criteria used to rule out variants. What does “sufficient evidence “ mean in this context? Unclear

L109 in the absence of further data, the evidence of KIF1A mutation being causative is not sufficient to support this phrasing

L116 I am unconvinced given the scant data provided to support this claim

L149 the use of multiple variant callers is not warranted if no information is provided as per the methods used to combine (potentially) contrasting SNP sets

L151 what are “in-house” procedures? Methods should be fully reproducible to asses the validity of the results.

Comments on the Quality of English Language

minor revision of English Language is required

Author Response

The manuscript by Bernard et al presents an interesting case in which KIF1A may be associated with an ALS phenotype. Although this perspective is intriguing, and in line with previous literature, I am not convinced that the data reported is sufficient to support the claims of the paper. In particular, the analysis of sequencing data is obscure, with many key steps in variant calling analysis and prioritization left undescribed: how to combine the results of multiple variant caller? Why to focus on this specific variant among the many variants potentially contributing to the ALS phenotype? As things stand, the authors identified the co-occurrence of the KIF1A variant with an ALS phenotype: this is far from a place where we could claim that this variant is indeed causative of the phenotype. The “in-house” method to prioritize variants is not acceptable in a peer review process. Finally, there is no effort in validating the variant via golden standard methods such as Sanger sequencing. Altogether, I am not convinced that the claim made in the paper is sufficiently demonstrated nor it could be demonstrated with additional revisions. Therefore, I am inclined to reject the submission.

Response: We acknowledge the need for a clearer description of our sequencing data analysis and variant prioritization process. We have updated the manuscript to include a detailed explanation of the variant calling and prioritization pipeline (see response to reviewer 1). Specifically, we combined results from multiple variant callers and focused on variants that met specific criteria, including rarity in population databases, predicted deleteriousness, and relevance to the patient's phenotype. Additionally, our bioinformatics pipeline is part of a national sequencing platform and follows a consensus used for numerous rare diseases under the France Genomic Medicine Plan. The term "in-house" was used inappropriately, and we have corrected this to reflect the standardized and widely accepted nature of our methodology.

Before proceeding with whole-genome sequencing, the patient was tested for mutations in the 40 known ALS-associated genes, including C9ORF72, which were all negative. This comprehensive pre-screening ruled out other known genetic causes of ALS. The KIF1A c.574A>G, p.(Ile192Val) variant was prioritized due to its de novo occurrence, absence in population databases, and localization in a mutational hot spot and critical and well-established functional domain. We have validated the KIF1A variant through Sanger sequencing, ensuring the accuracy of our findings. The details of this validation have been included in the revised manuscript and the corresponding figure 1. This step confirms the presence of the variant and addresses concerns about potential sequencing errors.

We hope these clarifications address the reviewer's concerns and demonstrate the robustness of our findings. We believe that the identification of the KIF1A variant in this patient significantly contributes to understanding the genetic landscape of ALS and warrants further investigation.

Line-by-line Comments:

L18: Remove words such as "Objectives", "Methods" etc., and opt for a narrative paragraph instead.

Response: We have restructured the abstract to present a narrative format without subheadings.

Revised Abstract: " We investigate the etiology of amyotrophic lateral sclerosis (ALS) in a 35-year-old woman presenting with progressive weakness in her left upper limb. Prior to sequencing, a comprehensive neurological work-up was performed, including neurological examination, electrophysiology, biomarker assessment, and brain and spinal cord MRI. Six months before evaluation, the patient experienced weakness and atrophy in her left hand, accompanied by brisk reflexes and Hoffman sign in the same arm. Electroneuromyography revealed lower motor neuron involvement in three body regions. Neurofilament light chains were elevated in her cerebrospinal fluid. Brain imaging showed asymmetrical T2 hyperintensity of the corticospinal tracts and T2 linear hypointensity of the precentral gyri. Trio genome sequencing identified a likely pathogenic de novo variant in the KIF1A gene (NM_001244008.2): c.574A>G, p.(Ile192Val). Pathogenic variants in KIF1A have been associated with a wide range of neurological manifestations called KIF1A-associated neurological diseases (KAND). This report describes a likely pathogenic de novo variant in KIF1A associated with ALS, expanding the phenotypic spectrum of KAND and our understanding of the pathophysiology of ALS."

L40: “number of genes”

Response: We have revised the text to clarify the number of genes and included a citation.

Revised text: " With the growing number of genes found to be associated with amyotrophic lateral sclerosis (ALS), which is currently approaching 40 [2], and the easier access to sequencing technologies, a pathogenic mutation is now identified in 70% of familial ALS cases [2] and up to 10% of sporadic ones [3]."

L40: the number of genes (40) requires a citation

Response: We have added the reference number to support the number of genes associated with ALS.

[2] " Feldman EL, Goutman SA, Petri S, et al. Amyotrophic lateral sclerosis. Lancet. 2022;400(10360):1363-1380. doi:10.1016/S0140-6736(22)01272-7."

L44: unclear, rephrase “absence of proven segregation”

Response: We have rephrased this sentence for clarity.

Revised text: " However, due to the typical late onset of the disease, interpreting rare gene variants in ALS can be challenging. This difficulty arises because there is often no opportunity to observe the segregation of the suspected variant within the family, either because the parents of the proband have passed away or because other family members are not available for testing. This situation can prevent the discovery of de novo variants."

L49: define HSP

Response: done

L54: define PEHO

Response: done

L57: downplay the sentence. “KIF1A variant found only in the patient”. Although unlikely, you can’t rule out a sequencing error unless a Sanger sequencing is performed.

Response: We have confirmed the KIF1A c.574A>G, p.(Ile192Val) variant by Sanger sequencing, ensuring the accuracy of our genomic findings. We have revised the manuscript to include this confirmation in the Results section. We added the following sentence: « The variant was confirmed by Sanger sequencing, ensuring the accuracy of the genomic findings. »

L61: why the sister was not included in the sequenced sample?

Response: The patient's sister chose not to undergo genetic testing at this time.

L70: refrain from using acronyms without defining them on first occurrence

Response: We have ensured that all acronyms are defined upon their first occurrence.

L86: there is no mention of the criteria used to rule out variants. What does “sufficient evidence “ mean in this context? Unclear

Response: We have clarified the criteria used to rule out variants.

Revised text: " Variant prioritization involved several key filtering steps. Initially, variants with a minor allele frequency greater than 1% in gnomAD were excluded. Subsequently, we filtered variants to retain only those affecting coding sequences or splice sites, and predicted to be deleterious by multiple in silico tools (e.g., for missense variants: PolyPhen-2, ClinPred, REVEL; for splicing variants: SpliceAI). Variants previously documented as pathogenic or likely pathogenic in ClinVar, as well as de novo or compound heterozygous variants, were prioritized. Finally, variants were evaluated in relation to the patient's phenotype and established gene-disease associations. "

L109: in the absence of further data, the evidence of KIF1A mutation being causative is not sufficient to support this phrasing

Response: We have adjusted the phrasing as following:

Revised Manuscript Text: "The present study supports the growing hypothesis that KIF1A is a gene associated with ALS, by identifying a probably pathogenic variant, classified as ACMG class 4, in a patient with ALS."

L116: I am unconvinced given the scant data provided to support this claim

Response: We believe that the evidence presented in our study is robust and adequately supports our claim. The identification of a de novo, likely pathogenic variant in KIF1A, supported by comprehensive clinical, electrophysiological, and imaging data, aligns with the growing body of literature associating KIF1A with neurological disorders, including ALS. Furthermore, our variant prioritization process, in accordance with established guidelines and validated through Sanger sequencing, strengthens the validity of our findings.

L149: the use of multiple variant callers is not warranted if no information is provided as per the methods used to combine (potentially) contrasting SNP sets

Response: The analysis pipeline utilized in our study involves the detection of single nucleotide variations (SNVs) and small insertions-deletions (indels up to 50bp) across the diagnostic target regions (canonical chromosomes 1 to 22, X, and Y) using the HaplotypeCaller tool from GATK. Variants genotyped in the index case were subsequently consolidated using GenotypeGVCFs from GATK, resulting in a unified variant call file per family. This approach ensures comprehensive variant detection within the specified genomic regions, addressing the reviewer's concern regarding the use of multiple variant callers without sufficient methodological explanation.

L151: what are “in-house” procedures? Methods should be fully reproducible to assess the validity of the results.

Response: We appreciate the reviewer’s request for more details on our variant prioritization pipeline. We have provided a brief description of the most important filtering steps used in our in-house procedures.

In the methods section: "Variant prioritization involved several key filtering steps. Initially, variants with a minor allele frequency greater than 1% in gnomAD were excluded. Subsequently, we filtered variants to retain only those affecting coding sequences or splice sites, and predicted to be deleterious by multiple in silico tools (e.g., for missense variants: PolyPhen-2, ClinPred, REVEL; for splicing variants: SpliceAI). Variants previously documented as pathogenic or likely pathogenic in ClinVar, as well as de novo or compound heterozygous variants, were prioritized. Finally, variants were evaluated in relation to the patient's phenotype and established gene-disease associations."

Reviewer 2 Report

Comments and Suggestions for Authors

In their case report titled "A Novel de Novo Missense Mutation in KIF1A Associated with Young-Onset Upper-Limb Amyotrophic Lateral Sclerosis" Bernard et al. present the case of a 35-year-old woman with amyotrophic lateral sclerosis. Trio-genome sequencing revealed a likely pathogenic KIF1A c.574A>G, p.(Ile192Val) variant. Pathogenic variants of KIF1A cause syndromic intellectual disability, but also hereditary spastic paraplegia and hereditary sensory and autonomic neuropathy. The case report expands the KIF1A associated neuronal diseases to amyotrophic lateral sclerosis and is of interest and in general well-presented. However, there are some points that require clarification before publication:

Major comments:

1. Whole-genome sequencing pipelines don't always assess STRs. The patient's slow disease course does not support C9orf72 HRE, but is should be nevertheless excluded as a potential cause. This can be done e.g. with ExpansionHunter from the PCR-free WGS data. 

2. The classification of KIF1A c.574A>G, p.(Ile192Val) as likely pathogenic seems justified, some people might argue for VUS. There are some details outside the ACMG classification criteria that make me question the variant's association with ALS. Namely, the observed variant is in the motor domain of the kinesin protein. Previously, KIF5A has been associated with ALS in addition to spastic paraplegia and Charcot-Marie-Tooth. However, the KIF5A ALS-associated variants were in the cargo-binding domain and spastic paraplegia/Charcot-Marie-Tooth associated variants in the motor domain. This discrepancy should be shortly discussed.

3. HGMD reports a variant in the same codon as disease mutation for intellectual disability (c.575 T>G, p.(I192S), https://pubmed.ncbi.nlm.nih.gov/35873028/). This I922S variant's REVEL score seems to be >0.8 whereas the score for Ile192Val is only 0.4. The prediction algorithms thus do not universally agree on the pathogenicity of the Ile192Val variant. Variants of the same codon can cause wildly different phenotypes, but it is not common. This phenotypic discepancy vs. the previous report could be shortly discussed.

Minor comments:

1. The authors could state which ACMG criteria they used for LP classificiation. Based on the wording I assume PS2+PM1+PM2.

2. I'd like more details on the variant priorisation pipeline than "in-house procedures": not all details are needed in a case-report but what were the most important filtering steps?

Author Response

Major Comments:

  1. Whole-genome sequencing pipelines don't always assess STRs. The patient's slow disease course does not support C9orf72 HRE, but it should be nevertheless excluded as a potential cause. This can be done e.g. with ExpansionHunter from the PCR-free WGS data.

Response: We excluded the presence of the pathogenic C9ORF72 HRE in our patient before proceeding with the whole-genome sequencing analysis. This information has been added to the manuscript for clarity.

Added text to the manuscript: "The absence of the pathogenic C9ORF72 hexanucleotide repeat expansion in our patient was confirmed prior to initiating whole-genome sequencing analysis."

  1. The classification of KIF1A c.574A>G, p.(Ile192Val) as likely pathogenic seems justified, but some people might argue for VUS. There are some details outside the ACMG classification criteria that make me question the variant's association with ALS. Namely, the observed variant is in the motor domain of the kinesin protein. Previously, KIF5A has been associated with ALS in addition to spastic paraplegia and Charcot-Marie-Tooth. However, the KIF5A ALS-associated variants were in the cargo-binding domain and spastic paraplegia/Charcot-Marie-Tooth associated variants in the motor domain. This discrepancy should be shortly discussed.

Response: We acknowledge the reviewer's insightful comment regarding the location of the variant in the motor domain of KIF1A. We agree that this is an important aspect to consider given the differences in domain-specific pathogenicity observed in KIF5A. We have added a discussion point to address this discrepancy and provide a more comprehensive interpretation of our findings.

Added text to the manuscript: " However, it is important to note that the KIF1A variant described here (c.574A>G, p.(Ile192Val)) is located in the motor domain of the kinesin protein. This contrasts with previously identified ALS-associated variants in KIF5A, which are found in the cargo-binding domain, leading to the formation of cytoplasmic aggregates, inclusions of mitochondria, and synaptic vesicles that are toxic to motor neurons through a gain-of-function mechanism [12-14] whereas spastic paraplegia and Charcot-Marie-Tooth-associated variants are located in the motor domain. This discrepancy suggests that while mutations in the motor domain of kinesin proteins can lead to a variety of neurological diseases, the specific domain affected may influence the phenotype."

  1. HGMD reports a variant in the same codon as disease mutation for intellectual disability (c.575 T>G, p.(I192S), https://pubmed.ncbi.nlm.nih.gov/35873028/). This I192S variant's REVEL score seems to be >0.8 whereas the score for Ile192Val is only 0.4. The prediction algorithms thus do not universally agree on the pathogenicity of the Ile192Val variant. Variants of the same codon can cause wildly different phenotypes, but it is not common. This phenotypic discrepancy vs. the previous report could be shortly discussed.

Response: We appreciate the reviewer bringing this to our attention. The presence of another pathogenic variant (I192S) at the same codon underscores the potential significance of this residue in KIF1A function. However, we acknowledge that the prediction algorithms do not universally agree on the pathogenicity of the Ile192Val variant. We have added a discussion point to address this phenotypic discrepancy and its implications.

Added text to the manuscript: " Further supporting the pathogenicity of the KIF1A c.574A>G variant, Human Gene Mutation Database (HGMD) reports a disease-causing mutation in the same codon (c.575 T>G, p.(Ile192Ser)), associated with intellectual disability. The REVEL score for p.(Ile192Ser) is >0.8, indicating a high likelihood of pathogenicity, while the score for p.(Ile192Val) is 0.4. Despite these differing predictions, the strong evolutionary conservation of isoleucine at position 192 across species and the damaging predictions from multiple in silico tools support the variant's likely pathogenic nature. "

Minor Comments:

  1. The authors could state which ACMG criteria they used for LP classification. Based on the wording I assume PS2+PM1+PM2.

Response: Thank you for this suggestion. We confirm that the classification of the KIF1A c.574A>G, p.(Ile192Val) variant as likely pathogenic (LP) was based on the ACMG criteria PS2, PM1, and PM2. We have added this information to the manuscript for clarity.

Added text to the manuscript: " According to the American College of Medical Genetics (ACMG) criteria [8], the KIF1A variant was classified as likely pathogenic (class 4). This classification was based on the following ACMG criteria: PS2 (de novo variant), PM1 (located in a mutational hot spot and/or critical and well-established functional domain), and PM2 (absent from controls in population databases). "

  1. I'd like more details on the variant prioritization pipeline than "in-house procedures": not all details are needed in a case-report but what were the most important filtering steps?

Response: We appreciate the reviewer’s request for more details on our variant prioritization pipeline. We have provided a brief description of the most important filtering steps used in our in-house procedures.

In the methods section: "Variant prioritization involved several key filtering steps. Initially, variants with a minor allele frequency greater than 1% in gnomAD were excluded. Subsequently, we filtered variants to retain only those affecting coding sequences or splice sites, and predicted to be deleterious by multiple in silico tools (e.g., for missense variants: PolyPhen-2, ClinPred, REVEL; for splicing variants: SpliceAI). Variants previously documented as pathogenic or likely pathogenic in ClinVar, as well as de novo or compound heterozygous variants, were prioritized. Finally, variants were evaluated in relation to the patient's phenotype and established gene-disease associations."

Round 2

Reviewer 2 Report

Comments and Suggestions for Authors

The authors have adequately addressed all my comments.